# Identification of Candidate Genes Controlling Red Seed Coat Color in Cowpea (*Vigna unguiculata* [L.] Walp)

Ira A. Herniter [1,2,*], María Muñoz-Amatriaín [3,*], Sassoum Lo [1,4], Yi-Ning Guo [1], Stefano Lonardi [5] and Timothy J. Close [1]

1  Department of Botany and Plant Sciences, University of California, Riverside, CA 92521, USA
2  Department of Plant Science, Rutgers University, New Brunswick, NJ 08901, USA
3  Departamento de Biología Molecular—Área de Genética, Universidad de León, 24071 León, Spain
4  Department of Plant Science, University of California, Davis, CA 95616, USA
5  Department of Computer Science and Engineering, University of California, Riverside, CA 92521, USA
*  Correspondence: ira.herniter@rutgers.edu (I.A.H.); mmuna@unileon.es (M.M.-A.)

**Abstract:** Seed coat color is an important consumer-related trait of the cowpea (*Vigna unguiculata* [L.] Walp.) and has been a subject of study for over a century. Utilizing newly available resources, including mapping populations, a high-density genotyping platform, and several genome assemblies, the red seed coat color has been mapped to two loci, *Red-1* (*R-1*) and *Red-2* (*R-2*), on Vu03 and Vu07, respectively. A gene model (*Vigun03g118700*) encoding a dihydroflavonol 4-reductase, a homolog of anthocyanidin reductase 1, which catalyzes the biosynthesis of epicatechin from cyanidin, has been identified as a candidate gene for *R-1*. Possible causative variants have also been identified for *Vigun03g118700*. A gene model on Vu07 (*Vigun07g118500*), with predicted nucleolar function and high relative expression in the developing seed, has been identified as a candidate for *R-2*. The observed red color is believed to be the result of a buildup of cyanidins in the seed coat.

**Keywords:** anthocyanidin reductase; cowpea; QTL analysis; seed coat color; SNP genotyping; *Vigna unguiculata*

## 1. Introduction

Cowpea (*Vigna unguiculata* [L.] Walp.) is a diploid (2n = 22) warm season legume that is primarily grown in Sub-Saharan Africa, where it serves as a major source of protein and calories. Additional production occurs in the Mediterranean Basin, Southeast Asia, Latin America, and the United States. Just over 8.9 million metric tons of dry cowpeas were reported to have been grown worldwide in 2019 [1], though these numbers do not include Brazil, Australia, and some other relatively large producers. Most of the production in Sub-Saharan Africa is carried out by smallholder farmers in marginal conditions, often as an intercrop with maize, sorghum, or millet [2]. Due to its high adaptability to both heat and drought and its association with nitrogen-fixing bacteria, cowpea is a versatile crop [2,3].

The most common form of consumption of cowpea is as a dry grain, with seeds used whole or ground into flour [4,5]. Seed coat color is an important consumer-related trait in cowpea, as consumers usually make decisions about the quality and presumed taste of a product based on appearance [6,7]. Cowpea grain displays a variety of colors, including black, brown, red, purple, and green. Each cowpea production region has preferred varieties, valuing certain color and pattern traits above others for determining quality and use. In West Africa, consumers pay a premium for seeds exhibiting certain characteristics specific to the locality, such as lack of color for use as flour or solid brown for use as whole beans [3,8–10]. In the United States, consumers prefer varieties with tight black eyes, commonly referred to as "black-eyed peas" [11].

Seed coat traits in cowpea have been studied since the early 20th century, when Spillman [12] and Harland [13], reviewed by Fery [14], explored the inheritance of factors controlling seed coat color and pattern. In a series of $F_2$ populations, Spillman [12] and Harland [13] identified the genetic factors responsible for color expression, including "Red" (*R*), which was confirmed by Saunders [15] and Drabo et al. [16]. The previous studies noted above, as well as similar studies of soybean (*Glycine max*) [17] and the common bean (*Phaseolus vulgaris*) [18–20], have shown that there are two loci involved in red coat color, each following a simple Mendelian inheritance. One of those loci seems closely associated with the color factor locus, which controls the overall pigmentation distribution on the seed coat [19–21]. This simple inheritance makes it relatively easy to develop reliable markers for use in marker-assisted breeding. Due to the high level of synteny between cowpea and other leguminous grain crops [22], markers developed in cowpea may be directly useful for breeding in closely related species.

A genotyping array for 51,128 single-nucleotide polymorphisms (SNPs) has been developed for cowpea [23] which, combined with relatively large populations, offers opportunities to improve the precision of genetic mapping. In particular, new populations have been developed for higher-resolution mapping including a Minicore collection representing a worldwide cross-section of cultivated cowpea ("UCR Minicore") [24] and an eight-parent multi-parent advanced generation inter-cross (MAGIC) population [25]. In addition, a reference genome sequence of cowpea (phytozome-next.jgi.doe.gov (accessed on 8 September 2023)) [22] and genome assemblies of six additional diverse accessions [26] (https://phytozome-next.jgi.doe.gov/cowpeapan/ (accessed on 8 September 2023)) have been produced recently. Here, we make use of the genetic and genomic resources available for the cowpea to map the red seed coat trait, determine candidate genes, and propose sources of the observed variation.

## 2. Materials and Methods

### 2.1. Plant Materials

Three cowpea populations were used for mapping: the UCR Minicore, which includes 368 accessions [24], a MAGIC population consisting of 305 lines [25], and an $F_2$ population developed as part of this work from a cross between Sasaque (Red, *r-1*) and Sanzi (Not Red, *R-1*), consisting of 108 individuals. Example images of the seed coats, including these two accessions, can be found in Figure 1.

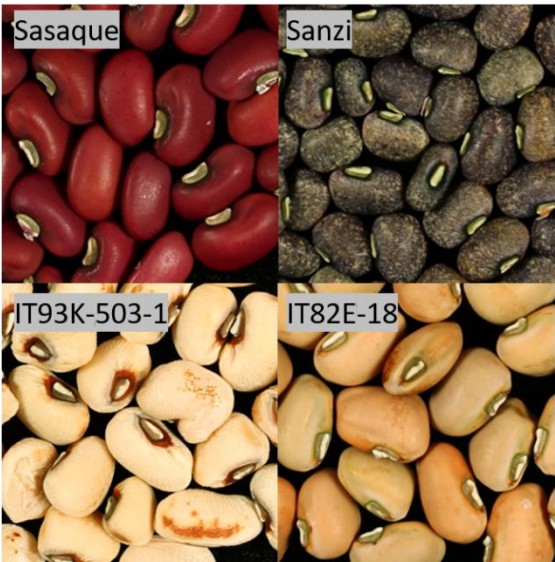

**Figure 1.** Images of accessions expressing and not expressing red seed coat color. Sasaque and IT93K-503-1 express red seed coat color; Sanzi and IT82E-18 do not.

Candidate gene sequences were compared using the sequences of the cowpea reference genome (IT97K-499-35-1) and the six additional sequences from the pan-genome (Sanzi, CB5, Suvita-2, UCR779, ZN016, and TZ30) [26].

These accessions, used as parents in the mapping populations and as part of the cowpea pan-genome [26], consist of at least one from each of the six identified subpopulations in global cowpea [24] (Supplementary Table S1).

### 2.2. SNP Genotyping and Data Curation

DNA was extracted from young leaf tissue using the Qiagen DNeasy Plant Mini Kit (Qiagen, Hilden, Germany) per the manufacturer's instructions. The Cowpea iSelect Consortium Array (Illumina Inc., San Diego, CA, USA), which assays 51,128 SNPs [23] was used to genotype each DNA sample. Genotyping was performed at the University of Southern California Molecular Genomics Core Facility (Los Angeles, CA, USA). The same custom cluster file used by Muñoz-Amatriaín et al. [23] was used for SNP calling.

For the UCR Minicore, SNPs with >20% missing data and a minor allele frequency (MAF) < 0.05 were eliminated, leaving a total of 42,603 SNPs for mapping. For the MAGIC population, SNP data and a genetic map were available from Huynh et al. [25], including 32,130 SNPs in 1568 genetic bins. For the Sasaque-by-Sanzi $F_2$ population, SNPs were filtered to remove non-polymorphic loci between the parents, leaving a total of 14,772 polymorphic markers (Supplementary Table S2).

### 2.3. Seed Coat Phenotyping

Phenotype data for seed coat traits were collected by visual examination of the seeds. The scored phenotypic classes consisted of "Red" and "Not Red", as dictated by the presence or absence of observable red pigmentation in the mature seed coat. For mapping purposes, the "Red" lines were marked with a "1" and the "Not Red" with a "0". Accessions with black seed coat pigmentation or no pigmentation whatsoever were regarded as missing data as it is believed that black pigmentation obscures other colors from view [27] and that the lack of color is controlled by the *Color Factor* locus Herniter et al. [21]. Phenotype data can be found in Supplementary Table S3 (UCR Minicore), Table S4 (MAGIC), and Table S5 (Sasaque-by-Sanzi).

Expected segregation ratios were determined based on the type of population and the parental and, when available, $F_1$ phenotypes. Expected segregation ratios were tested by chi-square analysis.

Dominance relationships were determined by examining the $F_1$ phenotypes of the Sasaque-by-Sanzi population, and segregating the parental phenotypes in the $F_2$ generation, as well as individuals from the early development of the MAGIC population. Seeds from relevant individuals were visually examined for the presence of red pigmentation on the seed coat.

### 2.4. Genetic Mapping of the Red Seed Coat Trait

Genetic mapping of the red seed coat trait was achieved with different methods for each type of population. In the MAGIC population, the R package "mpMap" [28] was used as described by Huynh et al. [25]. The significance cutoff values was determined through 1000 permutations, resulting in a threshold of $p = 8.10 \times 10^{-5}$ [$-\log10(p) = 4.09$]. Due to the high number of markers in the genotype data, markers spaced in 1 cM intervals were used.

A genome-wide association study (GWAS) was performed in the UCR Minicore to identify SNPs associated with the Red phenotype. The weighted mixed linear model (MLM) function [29] implemented in TASSEL v.5 [30] was used, with five principal components accounting for population structure in the dataset. The $-\log_{10}(p)$ values were plotted against the physical coordinates of the SNPs [26]. A Bonferroni correction was applied to correct multiple testing errors in GWAS, with the significance cut-off set at $\alpha/n$, where $\alpha$ is 0.05 and n is the number of tested markers, resulting in a cut-off of $-\log_{10}(p) = 5.93$.

For the Sasaque-by-Sanzi $F_2$ population, the genotype calls of each bulked DNA pool in the population were filtered to leave only the markers known to be polymorphic between the parents, and these were then sorted based on physical positions in the pseudochromosomes of IT97K-499-35 [26]. The genotype data were then examined visually in Microsoft Excel for areas where the recessive bulk was homozygous and the dominant bulk was heterozygous.

*2.5. Candidate Gene Identification and PCR Amplification*

The gene-annotated sequences of the overlapping QTL and region of interest for both Vu03 and Vu07 were obtained from the v1.2 of the reference genome sequence of cowpea (https://phytozome-next.jgi.doe.gov/info/Vunguiculata_v1_2 (accessed on 8 September 2023)) [22]. Candidate genes were identified by their location within the significant regions and their similarity to genes known to be involved in the red pigmentation of soybeans and commons bean in the literature (see Section 4). Additionally, data from the Gene Expression Atlas (VuGEA) of the reference genome was utilized (https://legumeinfo.org (accessed on 8 September 2023)) [31].

Primers were developed to amplify sections of the candidate gene *Vigun03g118700* in sasaque to examine the sequence for causative variations. Each primer pair, as well as the annealing temperatures, are listed in Supplementary Table S6. PCR was performed using the Thermo Scientific DreamTaq Green PCR Master Mix (Thermo Scientific, Waltham, MA, USA) per the manufacturer's instructions. Primers were developed using Primer3 v0.4.1 (bioinfo.ut.ee/primer3) and ordered from Integrated DNA Technology (Coralville, IA, USA). The PCR was run for 25–45 cycles with an annealing temperature compatible with the primer pair and an extension time of 60–75 s. Amplicons were confirmed by gel electrophoresis. PCR products of the expected size were purified using the QIAquick PCR Purification kit Catalog number 28104 (Qiagen, Hilden, Germany) and Sanger-sequenced in both directions on an Applied Biosystems 3730xl DNA Sequencer (University of California, Riverside IIGB Genomics Core).

Additional sequences of *Vigun03g118700* and *Vigun07g118600* were obtained from IT97K-499-35 (https://phytozome.jgi.doe.gov (accessed on 8 September 2023)) and the other genomes sequenced as part of the pan-genome, consisting of Sanzi (Red), CB5-2 (Not Red), Suvita-2 (Not Red), UCR779 (Not Red), TZ30 (Red), and ZN016 (Red) (https://phytozome-next.jgi.doe.gov/cowpeapan/ (accessed on 8 September 2023)) [26]. Complete nucleotide and amino acid sequences of both candidate genes were compared using MultAlin [32].

### 3. Results

*3.1. Phenotypic Variation for Red Seed Coat Color*

As described in Materials and Methods, phenotypic data were collected from the three populations and their parents (when applicable) by visual examination of the seeds. A summary of the phenotypic data, along with predicted segregation ratios, chi-square values, and probability can be found in Table 1, with demonstrations in Figure 2.

**Table 1.** Seed coat phenotypes and segregation ratios of the tested populations. "Other" includes seeds with black pigmentation or no pigmentation.

| Population (# of Lines) | Not Red | Red | Other | Pred. Seg. Ratio | $\chi^2$ | Probability |
|---|---|---|---|---|---|---|
| UCR Minicore (368) | 190 | 80 | 98 | -- | -- | -- |
| MAGIC (305) | 157 | 92 | 56 | 5:3 | 0.032 | 0.86 |
| Sasaque-by-Sanzi (108) | 86 | 22 | 0 | 3:1 | 1.23 | 0.27 |

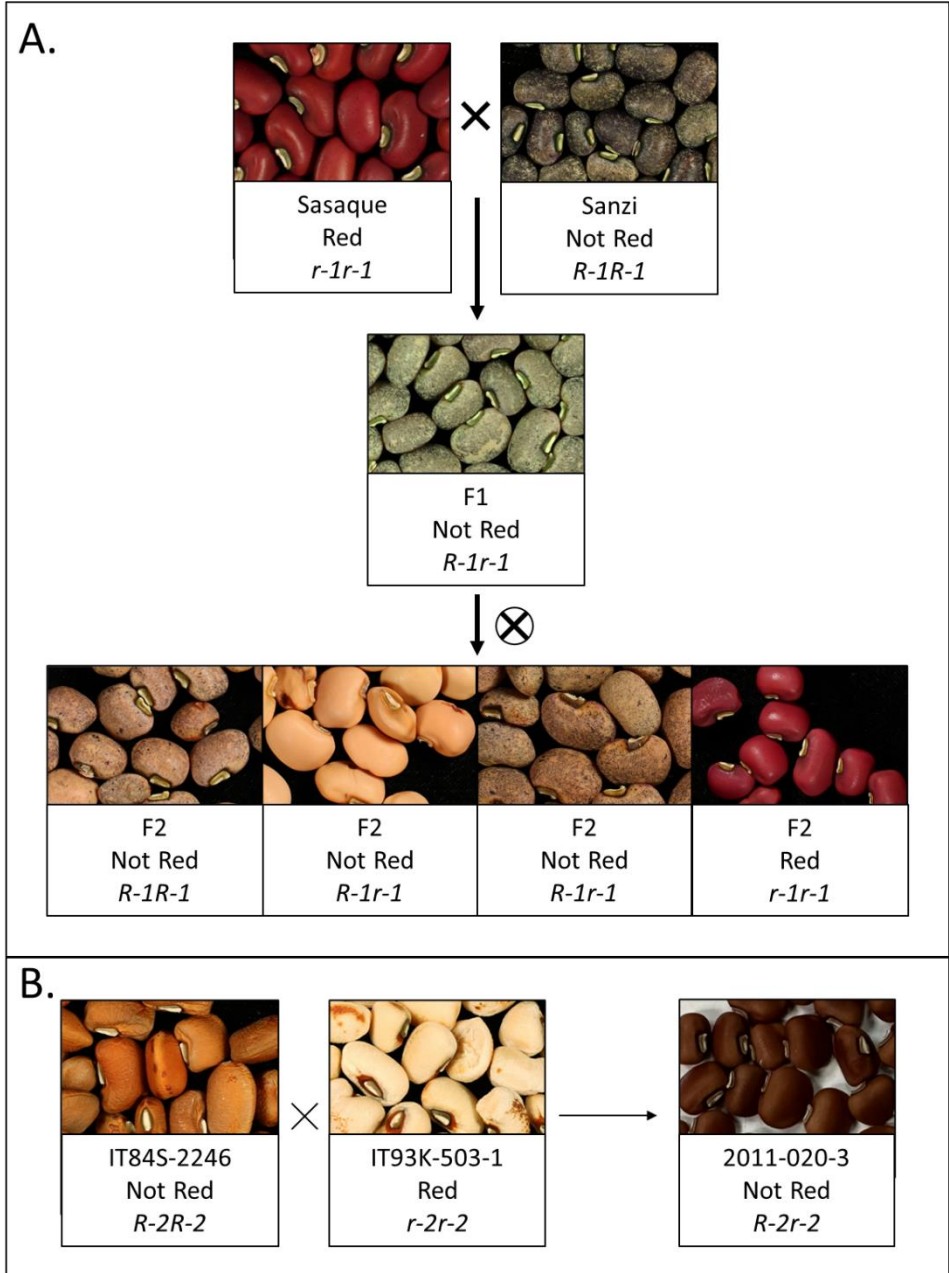

**Figure 2.** Segregation at the two *Red* loci, *R-1* and *R-2*. (**A**) Segregation at the *R-1* locus in the Sasaque-by-Sanzi F$_2$ population. (**B**) Cross between IT84S-2246 and IT93K-503-1 from the early development of the MAGIC population, resulting in Not Red phenotype in the seed coats of seeds of the F$_1$ maternal parent.

Expected segregation ratios were determined based on the type of population and the parental and F$_1$ phenotypes. For example, the Sasaque-by-Sanzi F$_2$ population was expected to segregate in a 3:1 ratio. For the MAGIC population, based on how the population was constructed [25], it was assumed that each fully homozygous parent had a roughly 1/8 chance to pass its genotype at a particular locus to a given RIL. Three of the MAGIC parents had a red seed coat (IT89KD-288, IT84S-2049, and IT93K-503-1) while the other five parents did not, resulting in an expected 5:3 segregation ratio.

The observed 3:1 segregation ratio of the Sasaque-by-Sanzi population together with the observed Not Red seed coat of the F$_1$ phenotype indicate the dominance of *R-1* (Not Red) over *r-1* (Red). Similarly, a cross from the early development of the MAGIC population demonstrated the dominance of *R-2* (Not Red) over *r-2* (Red): IT84S-2246 (*R-2R-2*, Not

Red) was crossed with IT93K-503-1 (*r-2r-2*, Red). The resulting plant, 2011-020-3 (*R-2r-2*), produced seeds with the Not Red phenotype (Figure 2B).

### 3.2. Loci Controlling Red Seed Coat Color

Two QTL, one each on Vu03 and Vu07, were identified using the three populations. The Vu03 QTL was identified in the UCR Minicore and the Sasaque × Sanzi F$_2$ population, while the Vu07 QTL was identified in the UCR Minicore and the MAGIC population (Table 2; Figure 3). Higher-resolution mapping for both genomic regions was achieved using the UCR Minicore due to the higher genetic diversity and number of recombination events available in this population. In particular, the significant region was 135 kbp in size for Vu03, and it was reduced to just one SNP position for Vu07. An analysis of linkage disequilibrium ($r^2$) between pairs of SNPs located 50 kbp upstream and downstream from the significant SNP position for Vu07 (2_02375) revealed low LD between the marker and nearby variants (analysis not shown). Complete information on the mapping results can be found in Supplementary Table S7 (UCR Minicore) and Supplementary Table S8 (MAGIC).

**Table 2.** Mapping results from the different populations. The base indicated in parentheses in the Effect column indicates the allele associated with the observed effect.

| Population | Chr | Peak SNP | Flanking SNPs | QTL pos. (bp) | LOD/−log10 (*p*) | R$^2$ (%) | Effect |
|---|---|---|---|---|---|---|---|
| Sasaque × Sanzi (F$_2$) | Vu03 | -- | 2_38494–2_05272 | 7,657,076–41,457,212 | -- | -- | |
| UCR Minicore | Vu03 | 2_20787 | 2_03389–2_20787 | 10,933,603–11,068,119 | 9.82 | 16.6 | −0.56 (G) |
| UCR Minicore | Vu07 | 2_02375 | -- | 22,014,760 | 6.26 | 9.8 | −0.61 (A) |
| MAGIC | Vu07 | -- | 2_51274–2_04829 | 14,976,909–22,928,810 | 29.81 | 38.82 | |

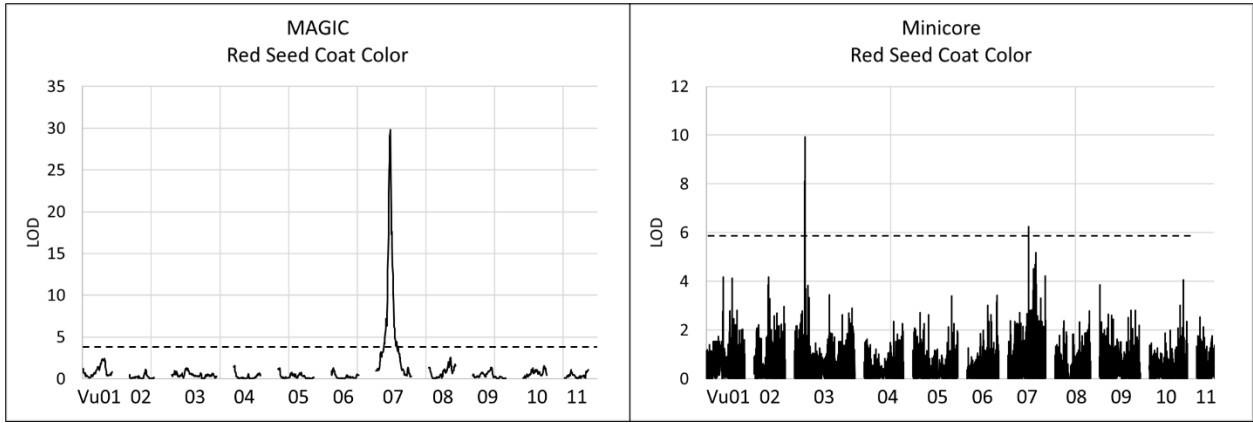

**Figure 3.** Mapping of the red seed coat trait in the MAGIC and UCR Minicore populations. The significance threshold is indicated by the dashed line.

### 3.3. Identification of Candidate Genes

The overlapping areas of the QTL and regions of interest on Vu03 and Vu07 were examined for candidate genes in the cowpea reference genome v1.2 (phytozome-next.jgi.doe.gov/info/Vunguiculata_v1_2, accessed on 8 September 2023) [22]. Eleven genes were identified in the minimal area (the area in which all the independent mappings identified) on Vu03 and one gene in the minimal area on Vu07 (Supplementary Table S9). Of the genes in the minimal area on Vu03, *Vigun03g118700* is the best candidate as it encodes a dihydroflavonol 4-reductase, which shows a highly elevated expression level during early seed development (Supplementary Figure S1) [31]. The peak SNP on Vu03 is also located within *Vigun03g118700*. The closest homologs to *Vigun03g118700* are anthocyanidin reductases identified in various leguminous plants, including *Vigna radiata*, *Vigna angularis*, common bean (*Phaseolus vulgaris*), and soybean (*Glycine max*), among others (Supplementary Table S10).

In the UCR Minicore, GWAS only identified a single significant SNP on Vu07, 2_02375, contained within the larger region identified in mapping the MAGIC population (Table 2, Supplementary Tables S7 and S8). This SNP is located in the third intron of *Vigun07g118600*, which is orthologous to the Arabidopsis *BAH1/NLA* gene encoding an E3 ubiquitin protein ligase with roles in defense response, plant growth, and development [33]. The significant SNP is not in high LD with nearby variants. *Vigun07g118600* does not have a high expression in the developing seed tissue of the reference genome cv. IT97K-499-35, which possesses black-eyed seeds (Supplementary Figure S2) [31].

To identify possible causative polymorphisms, sequence comparisons were made using the published cowpea genomes [26] and the sequence of *Vigun03g118700* amplified from the Sasaque genome. Variant locations in the following data are calculated from the start of the coding sequence.

In *Vigun03g118700*, several SNPs were observed among accessions, one of which could be causative (Table 3, Supplementary Figures S3 and S4). In the two Red-seeded genotypes from the pan-genome, TZ30 and ZN016, there was a substitution at base 593, C→A, resulting in an amino acid substitution, Ala→Asp. In Sasaque, there were two substitutions. One at base 661, T→G, resulting in an amino acid substitution, Met→Ser; the other at base 672, G→T, resulting in an amino acid substitution, Val→Phe.

**Table 3.** Observed variations in the coding DNA sequence of the candidate genes. The base number is calculated from the start of the coding sequence. Underlined bases in the AA code indicate SNP variants. "849+" indicates base 849 to the end of the coding sequence.

| **Vigun03g118700** | | | | | | |
|---|---|---|---|---|---|---|
| **BP** | **RefGen Base** | **Mutant Base** | **Genotypes Present In** | **AA Code** | **AA** | **Mutation Type** |
| 378 | C | T | UCR779 | AT$\underline{T}$→AT$\underline{C}$ | Ile→Ile | Silent |
| 592 | G | A | UCR779 | $\underline{A}$CT→$\underline{G}$CT | Thr→Ala | Polar Uncharged → hydrophobic |
| 593 | C | A | ZN016, TZ30 | G$\underline{C}$T→G$\underline{A}$T | Ala→Asp | hydrophobic → polar negative |
| 661 | T | G | Sasaque | AT$\underline{G}$→AG$\underline{C}$ | Met→Ser | hydrophobic → polar uncharged |
| 672 | G | T | Sasaque | $\underline{G}$TT→$\underline{T}$TT | Val→Phe | hydrophobic → hydrophobic |
| **Vigun07g118600** | | | | | | |
| **BP** | **RefGen Base** | **Mutant Base** | **Genotypes Present In** | **AA Code** | **AA** | **Mutation Type** |
| 761 | T | C | UCR779 | TT$\underline{C}$→TC$\underline{C}$ | Phe→Ser | hydrophobic → polar uncharged |
| 849+ | | | ZN016, TZ30 | | | Completely different AA sequence |

In addition, there were two substitutions in UCR779, a brown-seeded genotype from the pan-genome. At base 378, there is a silent mutation, C→T, still encoding Ile; and at base 592, G→A, resulting in an amino acid substitution, Thr-Ala.

In *Vigun07g118600*, additional mutations were observed (Table 3, Supplementary Figures S5 and S6). In both TZ30 and ZN016, there are major differences from base 850 onwards, resulting in a different amino acid sequence. In addition, at base 761 in UCR779, there is a SNP, T→C, resulting in an amino acid substitution.

## 4. Discussion

Previous studies, dating back over a hundred years, by Harland [13] and Spillman [12], reviewed by Fery [14] and confirmed by Drabo et al. [16], only identified the single locus *R* as being responsible for the red seed coat phenotype in cowpea. The present data suggests the presence of two loci, each of which can be independently responsible for the expression of red seed coat color. To avoid confusion, the locus on Vu03 is referred to as *R-1* while the locus on Vu07 is referred to as *R-2*. While previous studies on cowpea

and soybean have only identified a single locus responsible for the red seed coat color, Bassett et al. [34] identified two loci in the common bean, termed *Rk* (red kidney) and R (oxblood red). The candidate gene on Vu07, *Vigun07g118600*, is closely linked to the color factor locus, believed to be controlled by *Vigun07g110700* [21], and just 1.5 Mb [26] and 0.96 cM in distant according to the consensus map established by Muñoz-Amatriaín et al. [23]. This occurrence could explain why a second locus for the Red color in cowpea was not previously identified, as on seeds with only small areas of pigmentation, such as IT93K-503-1 (Figure 1) it can be difficult to discern Red (*r-1/r-1*, *r-2/r-2*) from Not Red (*R-1*, *R-2*). A similar linkage has been well-documented in the common bean (*Phaseolus vulgaris*) [34], and has been recently confirmed through trait mapping [19,20].

The flavonoid biosynthesis pathway has been well characterized in a number of plants, including soybean [17] and Arabidopsis [35], among others. An intermediate group of molecules in the flavonoid biosynthesis pathway are cyanidins. Cyanidins are well-described pigmentation molecules, identified in a wide range of plants, including red cabbage [36], clover [37], and various red berries [38], among others. Cyanidins appear red in low-pH (<3.5) conditions [39]. It has been well-established that much of the observed variation in plant organ pigmentation arises from differential levels of pigmentation molecules and pH in those organs [40].

The present study reports the identification of the *R-1* locus on Vu03. The most likely candidate gene, *Vigun03g118700*, encodes a dihydroflavonol 4-reductase protein (D4R). Based on the dominance pattern, with Not Red (*R-1*) dominant over Red (*r-1*), it is most likely that this is the previously identified *R* locus from the literature [14,16]. The identification of *Vigun03g118700* is given further credence by the independent mapping of red seed coat color. Fiscus et al. [41] performed genome-wide association studies using the IITA core collection, which consists of 2,082 cowpea accessions and which was similarly genotyped using the Cowpea iSelect Consortium Array. Using that population, the same SNP, 2_20787, was identified as being significantly correlated with red seed coat color.

A BLAST search of the soybean (*Glycine max*) genome on Phytozome, showed that the closest homolog of *Vigun03g118700* in soybean *Glyma.08G062000* (previously referred to as *Glyma08g06630*), with 90.63% identity and a e-value of 0.0 from pairwise BLAST of the coding sequence.

Kovinich et al. [17] demonstrated that soybean mutants with reduced function of the soybean *ANR1* gene (*Glyma.08G062000*) had a red seed coat phenotype. This would be consistent with a complete dominance model as a single functional copy of the gene would allow the successful catalysis of cyanidin to epicatechin and result in no red pigmentation, while an individual with two recessive alleles and nonfunctional D4R would not catalyze the biosynthesis of epicatechin, resulting in a buildup of cyanidin, and instead would express red seed coat pigmentation. The observed allele sequence in the common bean would match this, with the dominant *Rk* allele not expressing red pigmentation, the recessive *rk^d* allele expressing red pigmentation, and the semidominant *rk* allele in the middle with some red pigmentation [18].

Similarly, the closest homolog of *Vigun03g118700* in the common bean is *Phvul.002G218700*, with a 95.66% identity and an e-value of 0.0 from pairwise BLAST of the coding sequences. This gene is also predicted to encode an ANR. The recent mapping of color of the common bean by Campa et al. [19] and Sadohara et al. [20] identified SNPs which were significantly associated with the *a\** trait, which measures greenness and redness, as determined by the International Commission on Illumination [42]. Both found significantly correlated markers located 6–15 Mb up and downstream of *Phvul.002218700*. It is possible that the mapping did not identify markers closer because they did not map Red versus Not Red phenotypes directly, incorporating the green color as part of the phenotyping.

While the sequence comparisons of *Vigun03g118700* identified several potentially causative variations in Sasaque, TZ30, and ZN016, they also identified variations present in UCR779, which does not have any red pigmentation (Table 3). While the base substitution at base 378 (C→T, Ile→Ile) in *Vigun03g118700* is a silent mutation, the substitutions at base

592 (G→A, Thr→Ala) in *Vigun03118700* and at base 761 (T→C, Phe→Ser) in *Vigun07g118600* are more substantial. It is unclear why these mutations do not result in a similarly expected loss of the function to the mutations found in the Red phenotype varieties. Perhaps some insight could be gained from the determination that UCR779 is part of the most highly divergent subpopulation of cowpea [24], from eastern and southern Africa. It may be that its sequence represents alternative Not Red alleles (*r-1* & *R-2*). To better understand this, future studies should examine other diverse cowpea genotypes.

The *R-2* locus identified in the present study is mapped to Vu07. The Not Red color was dominant at this locus, similar to the *R* locus in the common bean (Figure 2B). Additionally, the *R-2* locus is closely linked to the color factor (*C*) locus, previously mapped by Herniter et al. [21]. The lone significant SNP identified here falls within an intron of *Vigun07g118600*, which is predicted to encode an E3 ubiquitin ligase. The available expression data for this gene shows quite low expression, with only about 0.5 TPM in the early developing seed and dropping off thereafter (Supplementary Figure S2). However, it should be noted that the only genotype for which transcription data are available is the cowpea reference genome, IT97K-499-35, which does not have red pigmentation and likely has the recessive, nonfunctional (*r-2*) allele, which is likely not expressed. This expression differential might be explained by variation in the promoter region. Future studies should seek to identify the transcription factor which controls the expression of *Vigun07g118600,* and they should seek to identify potential binding sites, which could vary between genotypes.

While the function of *Vigun03g118700* can be inferred based on previous research on other species, the function of *Vigun07g118600* in seed coat color is much less clear. Some possibilities include regulation of the flavonoid biosynthesis pathway, perhaps even exerting negative control over D4R (*R-1* locus, *Vigun03g118700*). Similarly, the closest homolog in soybean is *Glyma.10G018800*, with a 90.26% identity and an e-value of 0.0 from the pairwise BLAST of the coding sequence. Interestingly, *Glyma.10G018800* shows elevated expression in the root (https://phytozome-next.jgi.doe.gov/report/transcript/Gmax_Wm82_a4_v1/Glyma.10G018800.1 (accessed on 12 October 2023)) and has been identified as a candidate gene controlling root growth [43,44], potentially resulting from divergent evolution.

The closest homolog of *Vigun07g118600* in the common bean is *Phvul.007G143600*. In their mappings, both Sadohara et al. [20] and Campa et al. [19] identified a SNP 6 Mb downstream of *Phvul.007G143600*, as significantly associated with the *a\** trait. However, Sadohara et al. [20] did not identify candidate genes, and Campa et al. [19] did not consider this gene. It may be that, similarly to the function of the homolog in soybean, the function of *Phvul.007G143600* has diverged over evolutionary time. Future studies should examine the relationship between these homologs and their functional divergence.

**Supplementary Materials:** The following supplementary materials are available online at https://www.mdpi.com/article/10.3390/horticulturae10020161/s1, Figure S1: *Vigun03g118700* expression profile, Figure S2: *Vigun07g118600* expression profile, Figure S3: Coding sequence alignment of *Vigun03g118700* among the pan-genome sequences, Figure S4: Alignment of proposed amino acid sequences of Vigun03g118700 among genotypes in the pan-genome with the addition of Sasaque, Figure S5: Coding sequence alignment of *Vigun07g118600* among the pan-genome sequences, Figure S6: Alignment of proposed amino acid sequences of Vigun07g118600 among genotypes in the pan-genome, Table S1: Country of origin and subpopulation affinity of the population parents and pan-genome accessions, Table S2: Genotypes of the bulked samples from the $F_2$ population Sasaque-by-Sanzi, Table S3: Phenotypes of the UCR Minicore Collection, Table S4: Phenotypes of the UCR MAGIC population, Table S5: Phenotypes of the $F_2$ Sasaque-by-Sanzi population, Table S6: Primers used to amplify the candidate gene *Vigun03g118700* for sequencing, Table S7: GWAS results from the UCR Minicore Collection, Table S8: Mapping results from the UCR MAGIC population, Table S9: Candidate genes identified within the QTL peaks, Table S10: BLASTP Results of the *Vigun03g118700* transcript.

**Author Contributions:** Conceptualization, I.A.H. and T.J.C.; data curation, I.A.H. and S.L. (Stefano Lonardi); formal analysis, I.A.H. and M.M.-A.; funding acquisition, S.L. (Stefano Lonardi) and T.J.C.; investigation, I.A.H., M.M.-A., S.L. (Sassoum Lo) and Y.-N.G.; methodology, I.A.H.; project administration, I.A.H., M.M.-A. and T.J.C.; resources, I.A.H. and T.J.C.; software, S.L. (Stefano Lonardi); supervision, T.J.C.; validation, I.A.H. and M.M.-A.; visualization, I.A.H. and M.M.-A.; writing—original draft, I.A.H.; writing—review and editing, I.A.H., S.L. (Stefano Lonardi), M.M.-A., S.L. (Sassoum Lo), Y.-N.G. and T.J.C. All authors have read and agreed to the published version of the manuscript.

**Funding:** This study was supported by the Feed the Future Innovation Lab for Climate Resilient Cowpea (USAID Cooperative Agreement AID-OAA-A-13-00070), the National Science Foundation BREAD project "Advancing the Cowpea Genome for Food Security" (NSF IOS-1543963) and Hatch Project CA-R-BPS-5306-H.

**Data Availability Statement:** The original contributions presented in the study are included in the article and supplementary material; further inquiries can be directed to the corresponding author.

**Acknowledgments:** The authors thank Eric Castillo, Mia Rochford, Julia Valdepeña, and Sabrina Phengsy for assistance with seed photography; Steve Wanamaker for assistance in the analysis of the various genome sequences; Bao-Lam Huynh for providing the MAGIC population and attendant genotypic and pedigree information; Pei Xu and Xinyi Yu for providing images of the TZ30 and ZN016 genotypes.

**Conflicts of Interest:** The authors declare no conflicts of interest.

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
