# Peer review of "Identification of Candidate Genes Controlling Red Seed Coat Color in Cowpea (Vigna unguiculata [L.] Walp)"

_horticulturae, doi:10.3390/horticulturae10020161_

Round 1
Reviewer 1 Report
Comments and Suggestions for Authors
The manuscript “Identification of candidate genes controlling red seed coat color in cowpea (Vigna unguiculata [L.] Walp)” describes two candidate genes, Vigun03g118700 encoding a dihydroflavonol 4-reductase and Vigun07g118600 encoding an E3 ubiquitin ligase may be involved in red seed coat color. The findings is significant for the mechanism of controlling red seed coat in cowpea. However, some questions should be answered.
1. In Vigun03g118700, several SNPs were observed among accessions, one of which could be causative. Which one is causative?
2. The number of F2 generation of the Sasaque × Sanzi is small.
3. The Vigun03g118700 gene sequence in the cultivar Sasaque was mutated at 661 bp and 672 bp. Were the mutations detectrd in the red phenotype individuals of the F2 generation of the Sasaque × Sanzi at these two locations?
4. The format of the references is not uniform.
Comments on the Quality of English LanguageNA
Author Response
The manuscript “Identification of candidate genes controlling red seed coat color in cowpea (Vigna unguiculata [L.] Walp)” describes two candidate genes, Vigun03g118700 encoding a dihydroflavonol 4-reductase and Vigun07g118600 encoding an E3 ubiquitin ligase may be involved in red seed coat color. The findings is significant for the mechanism of controlling red seed coat in cowpea. However, some questions should be answered.
Thank you for your comments. We have responded to each comment below, with our responses in blue text.
- In Vigun03g118700, several SNPs were observed among accessions, one of which could be causative. Which one is causative?
The experiments performed for this paper cannot determine which, if any, of the observed mutations is causative. To determine that, directed mutagenesis or transformation would be required.
- The number of F2generation of the Sasaque × Sanzi is small.
The Sasaque x Sanzi population is relatively small compared to the other tested populations, but it is large enough to be confident in the observed segregation ratios. The X2 value of 1.23 with a probability of 0.27 (Table 1) indicates that the segregation ratio falls comfortably within predicted segregation ratio. Additionally, the population consists of plants grown from seeds collected off a single F1 plant.
- The Vigun03g118700 gene sequence in the cultivar Sasaque was mutated at 661 bp and 672 bp. Were the mutations detectrd in the red phenotype individuals of the F2generation of the Sasaque × Sanzi at these two locations?
We did not sequence the F2 plants. Given that we have the sequence of Vigun03g118700 from both parents, along with strong evidence from mapping and the literature we are confident in the identification of Vigun03g118700 as the candidate gene.
- The format of the references is not uniform.
The references have been reformatted.
Reviewer 2 Report
Comments and Suggestions for Authors
Two red seed coat colour candidate genes are identified through genetic mapping. However, they are from different lines Vu03 and Vu07. The Red-2 candidate gene is not expressed in the whole seed coat. The evidence is not strong enough to support the idea. It would be better to cross Vu03 and Vu07 to understand the patten in F2 population. Since they are recessive genes, this will provide the evidence of segregation and gene function. My comments are attached in the manuscript.

The writing in English is understandable and readable. but there are some sentences which are not written clearly, and it could be improved.
Author Response
Thank you for your comments. We have responded to each comment below, with our answers referring to the lines with the relevant comment.
Line 12: We have corrected the typo.
28: “Further” has been replaced with “Additional.”
106: “when available” has been removed. F1 phenotypes were used for both the MAGIC and F2 population.
108-112: “The” is required for proper grammar.
167: “when available” has been removed.
270, and elsewhere regarding the color of IT93K-503-1: IT93K-503-1 expresses red pigmentation in the seed coat. This is clearly visible in Figure 1 and Figure 2B. In IT93K-503-1, the red pigmentation is restricted to the small eye. It is not a white seed with a red eye, but a red seed with the pigment restricted to the eye. Where the pigmentation appears on the seed coat and the color of the pigmentation are separate. This is discussed in depth in Herniter et al 2019 (doi: 10.3389/fpls.2019.01346). The R-2 locus is very close to the C locus (~1.5Mb, 0.96 cM) but it is distinct.
Reviewer 3 Report
Comments and Suggestions for Authors
The study was aimed at identification of candidate genes controlling red seed coat color in cowpea (Vigna unguiculata [L.] Walp). The Authors found that the gene model (Vigun03g118700) encoding a dihydroflavonol 4-reductase, a homolog of anthocyanidin reductase 1, which catalyzes the biosynthesis of epicatechin from cyanidin, has been identified as a candidate gene for R-1. Possible causative variants have been also identified for Vigun03g118700. The gene model on Vu07 (Vigun07g118500), with predicted nucleolar function and high relative expression in the developing seed, has been identified as a candidate for R-2.
In my opinion, the research topic is interesting, the used methods, presentation of the results and their results are generally correct. However, I recommend only some minor improvements:
- I suggest removing the Figure 4. (Alignment of proposed amino acid sequences of Vigun03g118700 among genotypes in the pangenome with the addition of Sasaque) into the Supplementary file.
- Similarly, the Figure 5. (Alignment of proposed amino acid sequences of Vigun07g118600 among genotypes in the pangenome) should be removed into the Supplementary file.
- Although a large number of research results are presented, the discussion is not adequately in-depth and multifaceted. Hence, I recommend re-writing the discussion in the direction of a broader interpretation of the results.
- Conclusion part should be re-written into a more relevant form (not repeating the aim or the results of the study).
Author Response
The study was aimed at identification of candidate genes controlling red seed coat color in cowpea (Vigna unguiculata [L.] Walp). The Authors found that the gene model (Vigun03g118700) encoding a dihydroflavonol 4-reductase, a homolog of anthocyanidin reductase 1, which catalyzes the biosynthesis of epicatechin from cyanidin, has been identified as a candidate gene for R-1. Possible causative variants have been also identified for Vigun03g118700. The gene model on Vu07 (Vigun07g118500), with predicted nucleolar function and high relative expression in the developing seed, has been identified as a candidate for R-2.
Thank you for your comments. We have responded to each comment below, with our responses in blue text.
In my opinion, the research topic is interesting, the used methods, presentation of the results and their results are generally correct. However, I recommend only some minor improvements:
- I suggest removing the Figure 4. (Alignment of proposed amino acid sequences of Vigun03g118700 among genotypes in the pangenome with the addition of Sasaque) into the Supplementary file.
We agree. The figure has been moved to the supplementary file (Supplementary Figure S4).
- Similarly, the Figure 5. (Alignment of proposed amino acid sequences of Vigun07g118600 among genotypes in the pangenome) should be removed into the Supplementary file.
We agree. The figure has been moved to the supplementary file (Supplementary Figure S6).
- Although a large number of research results are presented, the discussion is not adequately in-depth and multifaceted. Hence, I recommend re-writing the discussion in the direction of a broader interpretation of the results.
We believe the discussion goes into enough depth regarding the history of seed coat studies in cowpea, how red color loci relate to other previously identified loci in cowpea and related species, and what future directions for research would be. Please let us know if there is a specific topic you would like addressed in the discussion and we will be happy to include it.
- Conclusion part should be re-written into a more relevant form (not repeating the aim or the results of the study).
We have removed the conclusion section.